# Optical Amplification in Hollow-Core Negative-Curvature Fibers Doped with Perovskite CsPbBr_3_ Nanocrystals

**DOI:** 10.3390/nano9060868

**Published:** 2019-06-07

**Authors:** Juan Navarro-Arenas, Isaac Suárez, Juan P. Martínez-Pastor, Albert Ferrando, Andrés F. Gualdrón-Reyes, Iván Mora-Seró, Shou-Fei Gao, Ying-Ying Wang, Pu Wang, Zhipei Sun

**Affiliations:** 1Instituto de Ciencia de Materiales (ICMUV), Universidad de Valencia, C/Catedrático José Beltrán, 2, E-46980 Paterna, Spain; isaac.suarez@urjc.es; 2Escuela de Ingenieros de Telecomunicación, Universidad Rey Juan Carlos, Camino del Molino s/n E 28942 Fuenlabrada, Spain; 3Departament d’Òptica i Optometria i Ciències de la Visió, Universitat de València, Dr Moliner, 50, 46100 Burjassot, Valencia, Spain; albert.ferrando@uv.es; 4Institute of Advanced Materials (INAM), University Jaume I, Avenida de Vicent Sos Baynat, s/n, 12006 Castelló de la Plana, Castellón, Spain; andres2127240@correo.uis.edu.co (A.F.G.-R.); sero@uji.es (I.M.-S.); 5Biofuels Lab-IBEAR, Faculty of Basic Sciences, University of Pamplona, 543050 Pamplona, Colombia; 6Beijing Engineering Research Centre of Laser Technology, Institute of Laser Engineering, Beijing University of Technology, 100124 Beijing, China; gaofei@shanghaitech.edu.cn (S.-F.G.); wangyingying@bjut.edu.cn (Y.-Y.W.); wangpuemail@googlemail.com (P.W.); 7Department of Electronics and Nanoengineering, Aalto University, Tietotie 3, 02150 Espoo, Finland; zhipei.sun@aalto.fi; 8QTF Centre of Excellence, Department of Applied Physics, Aalto University, FI-00076 Aalto, Finland

**Keywords:** hollow core fibers, perovskite nanocrystals, signal regeneration/amplification, nonlinear optical properties

## Abstract

We report a hollow-core negative-curvature fiber (HC-NCF) optical signal amplifier fabricated by the filling of the air microchannels of the fiber with all-inorganic CsPbBr_3_ perovskite nanocrystals (PNCs). The optimum fabrication conditions were found to enhance the optical gain, up to +3 dB in the best device. Experimental results were approximately reproduced by a gain assisted mechanism based on the nonlinear optical properties of the PNCs, indicating that signal regeneration can be achieved under low pump powers, much below the threshold of stimulated emission. The results can pave the road for new functionalities of the HC-NCF with PNCs, such as optical amplification, nonlinear frequency conversion and gas sensors.

## 1. Introduction

*Hollow-core negative-curvature fibers* (HC-NCF) are a special class of optical fibers where the cladding is composed by an array of hollow glass tubes surrounding a hollow-core [1]. HC-NCFs are also referred to *hypocycloid-shaped hollow-core fibers* [2], *revolver fibers* [3], *tube lattice fibers* [4] or *hollow-core anti-resonant fibers* [5,6]. In all these fibers, the curvature of the normal surface in the region surrounding the central hollow-core is negative with respect to the radial unit vector. HC-NCFs allow for anti-resonant reflecting optical waveguiding mechanism [7], and differ from *hollow-core photonic crystal fibers* (HC-PCF), formed by a two-dimensional periodic cladding structure [8]. Conversely, the degrees of freedom introduced by the core-shape boundary in HC-NCFs allow overcoming light leakage [9]. In this way, since their appearance in 2002, with the Kagome-type broadband hollow-core fiber [10], the HC-NCFs have attracted significant attention of the photonics community, and, as such, fibers have the potential to reach new frontiers in the fiber optics field and serve as a workbench for developing new technologies. These fibers support extremely large transmission bandwidths and low field dielectric overlap, avoiding a significant decrease of absorption losses with respect to other fiber designs and the presence of nonlinearities such as the Kerr effect, making them excellent waveguides for high pulse delivery and other high optical density applications [11]. Examples of practical applications include the development of high-power diode-pumped alkali lasers [12] or the signal splitting in optical telecommunications networks [13,14].

Hollow-core fibers can also serve as hosts, working as a *lab-on-a-fiber* for the study of optical properties of materials. For example, filling air holes with different materials that have different functionalities allows a large range of potential applications, such as chemo-/bio-sensing [15]. Given the relative youth of this kind of microstructured fibers, only few research works have been developed in this direction, such as optofluidic lasers [16] and sensors [17]. Thus far, the combination of emitting materials with HC-NCFs has not been examined, although we can find a few HC-PCF demonstrations of lasing at visible wavelengths by using CdZnS/ZnS core–shell colloidal Quantum Dots (QDs) [18,19,20], and optical amplification at telecommunication wavelengths with PbS colloidal QDs [21,22].

In this context, all-inorganic CsPbX_3_ (X = Cl, Br, I) perovskite nanocrystals (PNCs) have emerged as efficient active materials for optoelectronics [23]. As colloidal QDs, colloidal PNCs are synthetized under low cost solution process chemistry (hot-injection method), and demonstrate a high efficiency of light absorption and emission with a tunable band gap dependent on the composition [24]. As an advantage with colloidal QDs, the photoluminescence (PL) of PNCs is characterized by narrower spectral lines (10–20 nm, depending on temperature) [25] and smaller pump thresholds for optical gain [26]. Indeed, since the first publication in 2015 [25], PNCs have been extensively studied in different fields with demonstrated applications in LEDs [27], lasing [28,29,30] and optical amplification [31].

In this paper, we present the inclusion of CsPbBr_3_ PNCs in HC-NCFs as potential all-fiber active optical microdevices. For this purpose, the air holes of the micro-structured silica fiber were filled with colloidal PNCs by capillary forces. The method for infiltrating PNCs into the HC-NCF was properly optimized to obtain a homogeneous adhesion of the nanocrystals to their walls along tens of centimeters. In the resulting active photonic structure that was modeled by COMSOL Multiphysics software, a probe beam coupled at the input facet of the HC-NCF was enhanced up to +3 dB under 405 mm Continuous Wave (CW) optical pumping. Although this optical amplification can be phenomenologically modeled by the theory of Erbium-doped fiber amplifiers [32], stimulated emission conditions were not reached under the low power CW laser pumping used in our experiments. Alternatively, we suggest a nonlinear mechanism to explain the experimentally observed amplification, even if further investigation is needed to elucidate the ultimate origin for this mechanism.

These results not only represent a thresholdless gain assisted mechanism for HC-NCFs, but also pave the road of integrating a broad range of functionalities within these fibers. In particular, it is well known that nonlinear processes are necessary to develop a full optical signal processing in a telecommunication system [33]. In this way, the nonlinear mechanism proposed here could be useful in the implementation of different tasks, such as signal regeneration, modulation, multiplexing or demultiplexing, among others [33]. In addition, the particular structure of HC-NFCs results in an excellent platform for high-toxicity gases sensing [34]. In these conditions, since PNCs also demonstrate a high sensitivity to different gas or liquid compounds [35], the insertion of PNCs into the HC-NFCs is expected to allow a stronger interaction with these gases and hence the basis to develop high throughput optical sensors.

## 2. Experimental Details

### 2.1. CsPbBr_3_ Nanocrystal Synthesis

CsPbBr_3_ nanocrystals were synthetized following the hot-injection method [25] with some modifications. All the reactants were used as received without additional purification process. Briefly, a Cs-oleate solution was prepared by mixing 0.41 g Cs_2_CO_3_ (Sigma-Aldrich, Madrid, Spain, 99.9%), 1.25 mL of oleic acid (OA, Sigma Aldrich, Spain, 90%) and 20 mL of 1-octadecene (1-ODE, Sigma-Aldrich, Spain, 90%) into a 50 mL three-neck flask at 120 °C under vacuum for 1 h, stirring constantly. Then, the mixture was N_2_-purged and heated at 150 °C until the Cs_2_CO_3_ was completely dissolved. The solution was stored under N_2_, keeping the temperature at 100 °C to prevent the Cs-oleate precipitation. For the synthesis of CsPbBr_3_ nanocrystals, 0.69 g PbBr_2_ (ABCR, Spain, 99.999%) was mixed with 50 mL of 1-ODE into a 100 mL three-neck flask. The mixture was heated at 120 °C under vacuum for 1 h, keeping a constant stirring. Then, 5 mL of both OA and oleylamine (OLA, Sigma-Aldrich, Spain, 98%) were separately added to the flask under N_2_ atmosphere, and rapidly heated to reach 170 °C, injecting quickly 4 mL of Cs-oleate solution. Lastly, the flask was immersed into a bath ice for 5 s to quench the reaction mixture. For the isolation of PNCs, the colloidal solutions were centrifuged at 4700 rpm for 10 min. The PNCs were separated after discarding the supernatant and re-dispersed in hexane to prepare a colloidal solution with a concentration of 50 mg/mL. The absorption spectrum of this colloidal solution exhibits a well-defined excitonic edge around 505 nm, whereas the PL spectrum is rather narrow (24.8 nm) and centered at 514 nm (see Figure 1a). Absorption/PL data are in agreement with previously published results for cubic-shape PNCs of similar size (10 nm in average, as shown in Figure 1b) [25].

### 2.2. Fabrication and Characteristics of the HC-NCF

The Hollow-Core fiber was fabricated by the stack-and-draw technique. This fiber is quite similar with the one published elsewhere [36]: a HC fiber with seven untouched capillary tubes built around the hollow-core. The HC-NCF was properly designed to span the transmission window between 400 and 780 nm to present low attenuation at both the emission wavelength of the PNCs (515 nm) and the pump beam wavelength (405 nm). Indeed, propagation losses of the fundamental air mode at the pass band are reduced down to 0.04 dB·cm^−1^ and 0.14 dB·cm^−1^ at 514 nm and 405 nm, respectively. The fiber was loaded with PNCs following a procedure described in Section 3. Finally, fibers where cleaved with a fiber cleaver (model CT-101, Fujikura, Japan). An optical microscope (WITec alpha300, Germany) was used to map the backscattered PL light of the PNCs infiltrated in the HC-NCF.

### 2.3. Optical Characterization Setup

Optical gain measurements were carried out with the experimental set-up shown in Figure 2 at room temperature and ambient conditions. It consisted of two lensed fibers positioned at the input and output ends of the HC-NCF by computer-controlled XYZ piezo translators with micrometric precision. In addition, both tipped fiber couplers were placed on a rotating head with two additional degrees of freedom. A focused fiber coupler (FC) launched both pump (405 nm) and probe signal (515 nm) beams inside the input fiber, while the other fiber tip was coupled to the output end and connected to a spectrometer (Ocean Optics 2000) through a 450 nm long-pass (LP) filter to cut the pump light in the spectrometer. Intensity of the pump beam was externally controlled up to 3 µW, while probe signal intensity was reduced down to 1 nW.

## 3. Fabrication of HC-NCFs Doped with CsPbBr_3_ Nanocrystals

A short piece of HC-NCF was submerged inside the colloidal solution of PNCs to load the fiber’s air tubes (of radii 3 μm and radially distributed with a period of 10 μm beneath the cladding of the HC-NCF, as shown in Figure 3a) with PNCs by capillary action. Since the size of the PNCs (10 nm) was much smaller than these holes, the colloidal solution easily infiltrated through the SiO_2_–air hole walls, where PNCs were deposited, as demonstrated below. The length, *h*, covered by the liquid inside the air capillary can be calculated by applying the Jurin’s law, which relates the radius *r* of the fiber microtube with the surface tension *γ* of the PNC-colloidal solution [37]:(1)$$h=\frac{2\gamma cos\left(\Theta \right)}{\rho gr}$$
where *g* is the gravitational acceleration, *ρ* is the liquid density and *Θ* is the contact angle between the liquid and the capillary wall. Since the concentration of PNCs in the solvent was relatively low (see below), *γ* and *ρ* can be approximated by the values reported for hexane, 18 dyn/cm and 0.659 g/mL [38], respectively. In these conditions, and assuming a contact angle of *Θ* = 0 (to estimate the upper limit for *h*), Jurin’s law predicts that h is much longer than used fiber lengths (0.6–1.4 cm). The optimum concentration of the colloidal solution containing the CsPbBr_3_ PNCs was around 1 mg/mL, being the dipping process maintained over one day to get a uniform loading of the HC-NCF. A careful cleaving procedure was carried out to smooth the fiber facets before characterization and gain experiments.

As expected, PNC-loaded HC-NCFs show a uniform PL along 140 mm, as shown in Figure 3b (image of the whole 140 mm long fiber and zoom of a 0.8 mm long fragment). In addition, PL cross section images measured at the input (Figure 3c) and middle (Figure 3d) faces of the PNC-loaded HC-NCF clearly demonstrate that light emission is concentrated in the air tube walls and maintained along the fiber (also visible in Figure 3b).

The HC-NCF was simulated with COMSOL Multiphysics software (version 5.3, Sweden) by using the finite element method with a circular Perfectly Matched Layer (PML) applied to the outermost ring of the structure geometry (see Figure 3e,f). First, simulation of the unloaded membrane showed the central HE_11_-core mode (this mode is depicted in Figure 3e) with an effective index of 0.99984 reported from this type of fibers [36]. However, the adhesion of the PNCs on both sides of the walls of the fiber allows the propagation of modes through the walls of the HC-NCF. The PNCs are considered to be attached to both sides of the walls. In particular, seven degenerated cladding modes are supported with an effective index of 1.4574 (Figure 3f). Indeed, the calculated modal distributions corroborate the experimental PL-map shown in Figure 3c,d. These simulations were carried out by considering seven non-touching SiO_2_ capillary membranes with a thickness of 300 nm surrounded by two coating layers (the inner and the outer) of PNCs with a thickness of 55 nm. The PNC film thickness was estimated from the optical images (Figure 3c,d) and contrasted with profilometry measurements from dip-coated PNC films prepared on top of a SiO_2_ substrate. In both cases, the thickness was carefully controlled by the right concentration of PNCs in the solvent (from 1 to 50 mg/mL). In addition, the uniform deposition of PNCs along the fiber was checked by sampling the cross section of the loaded HC-NCF device at different lengths (1, 2 and 3 cm from one of the open ends), obtaining a mean thickness of around 60 ± 10 nm. The operation wavelength was set equal to 500 nm (since there is not too much difference in the 400–550 nm range), the refractive index of membrane was fixed to 1.46 (SiO_2_), and the refractive index of the PNCs was considered in the range of *n*_p_ = 1.8–2.2, as reported elsewhere [31].

Therefore, the adhesion of PNCs strongly modifies the light propagation properties of the HC-NCF for both pumping and signal laser beams. Indeed, the super-mode distribution depends on the thickness of PNCs deposited at the capillary walls. For this purpose, we evaluated the mode confinement of the super-mode as a function of the PNC layer thickness through the following overlap integral $\Gamma_{p}$:(2)$$\Gamma_{p}=\frac{\iint_{PNC\;Rings}\left|\langle S_{z}\left(x,y\right)\rangle \right|dxdy}{\iint_{-\infty }^{\infty }\left|S_{z}\left(x,y\right)\right|dxdy}$$

Here, the modulus of the time-averaged normal component of the Poynting vector is integrated in the area occupied by the nanoparticles and compared to the whole extension of the modal field distribution. In particular, $\Gamma_{p}$ = 0.2 is calculated from a thickness of 55 nm for the PNC layer, as determined from the optical images.

## 4. Results and Discussion

### 4.1. Optical Amplification in the HC-NCF Doped with PNCs

Optical amplification characterization in the PNC-doped HC-NCFs was carried out by coupling pump and probe signal beams on the input facet of the HC-NCF, as commonly used in erbium-doped fiber optical amplifiers [32] and described in Section 2.3 (Figure 2). In these conditions, optical amplification can be observed as an enhancement of the probe signal by increasing pump power. Experimental spectra recorded for different excitation fluencies (data symbols in Figure 4a) were deconvoluted by the sum of two Gaussian functions (solid lines in Figure 4a) to study the propagation of the PL generated by the CW-laser pump (405 nm) and the probe signal light (515 nm), independently. On the one hand, the integrated PL of PNCs exhibits a sharp increase under low excitation regime (i.e., < 1 µW of laser pump power) and PL saturates above this value (brown solid circles in Figure 4b), whereas the probe signal increases linearly in the entire range of laser pump power (green solid circles in Figure 4b). Here, since the excitation light coupled in the cladding modes will be strongly attenuated by the absorption of the PNCs, the pump beam is considered to be propagated through the fundamental air mode. In these conditions, the observed saturation effect can be attributed to propagation losses of the pump beam along the HC-NCF core (0.14 dB/cm). Obviously, these losses are quite low and the propagation of the pump beam is assured through the entire length of the fiber, hence being effective the excitation of PL from PNCs by the evanescent field of the HC-NCF HE_11_-core mode (Figure 3e). Indeed, a similar co-propagation of pump and PL was already exploited in planar waveguide amplifiers [26,39]. From these experiments, we deduce a signal amplification or gain P_out_/P_in_ (ratio of probe signal at the output facet of the fiber over that at the input one) that increases with the laser pump power and with the HC-NCF length (Figure 4c). Consequently, the probe signal might well be enhanced by the pump-activated amplification, due to the interaction with the infiltrated PNCs in the fiber.

The full gain of the HC-NCF can be calculated by:(3)$$G_{dB}=10\log_{10}\left(\frac{I_{Probe}\left(P_{a}\right)}{I_{Probe}\left(0\right)}\right)$$
where *I_probe_(P_a_)* is the output signal intensity at the probe light wavelength as a function of the absorbed pump power (*P_a_*). In this way, a signal amplification *P_out_/P_in_* = 2 (3 dB gain) is demonstrated for the highest pump power in a 140 mm long HC-NCF (black solid squares in Figure 4c). The optical amplification is significantly reduced for shorter fibers, 80 and 60 mm long, where less than 2 and 1 dB is measured (blue and red solid squares in Figure 4c), respectively. The observed optical amplification as a function of the HC-NCF length follows the *e^(g·L)^* law commonly observed in waveguide or fiber optical amplifiers [40]. The saturation observed in P_out_/P_in_, especially for short fibers, can be explained by the attenuation of the pump beam, as explained above. It is worth noting that these three HC-NCFs in Figure 4c were doped by using the same PNC-solution with a concentration of 1 mg/mL, which is the optimum value for the fabrication process described in Section 3. A minimum amount of PNCs deposited in the HC-NCF are required, in agreement with previous results of waveguides doped with other colloidal nanocrystals or dyes [39].

### 4.2. Modelling Optical Amplification.

The optical gain generated in a waveguide is usually calculated by a rate equation model coupled with the propagation equation [41]. Here, an analytical solution of the power propagated in the HC-NCF can be obtained by using the formalism developed in erbium-doped fiber amplifiers, applicable under small-signal (<20 dB) amplification [42]. This model reproduces the signal and gain generated in an optical amplifier with amplification of the spontaneous emission (ASE). The discussion is limited to a signal beam (s) and a pump beam (p), identically treated in the formalism. In addition, the pump and the PNC-doping concentration are considered homogeneous over the cross-section area *A* of the core, whereas the propagation losses for the signal beam are considered negligible. In these conditions, the gain of the signal beam obeys the analytical equation derived elsewhere [43]:(4)$$G_{s}\left(L\right)=10\;\log_{10}\left(e\right)\left[\frac{\Gamma_{p}\left(\sigma_{s}^{e}+\sigma_{s}^{a}\right)\tau \phi_{p}}{h\nu_{p}A}P_{a}-\alpha_{s}L\right]dB$$
where $\alpha_{s}=\varrho \Gamma_{s}\sigma_{s}^{a}$ is the signal absorption constant and the saturation power, ϱ is the concentration of PNCs per unit volume, τ is the radiative lifetime of the PNCs, $\sigma_{s}^{a}$ and $\sigma_{s}^{e}$ are the absorption and emission cross sections for the signal beam, $\Gamma_{p}$ is the modal confinement factor calculated for the pump (or the signal beam) through Equation (2), *P_a_* the absorbed laser pumping power by PNCs.

Here, the area *A* was set 7.5 µm^2^, τ was fixed to τ = 15 ns as reported elsewhere for this kind of nanocrystals [28], *Γ_p_* = 0.2 was obtained from the COMSOL simulation presented in Section 3, and $\nu_{p}$ is the pump frequency. The quantum efficiency of the emitters was set to be around $\phi_{p}$ = 0.85, as reported for these PNCs [25]. Although it is well known that this parameter decreases in thin films [44], it does not have a critical impact in our simulation results, and can range between 0.5 and 0.85, as we measured for a previous publication.

In these conditions, absorption and emission cross sections can be deduced by fitting to Equation (4) the experimental results presented in Figure 4d (from data in Figure 4c) for the 140 mm long PNC-doped HC-NCF (blue hollow squares). For this purpose, $\sigma^{a}$ and $\sigma^{e}$ can be related with McAmber theory [45] by $\sigma^{e}=\sigma^{a}\exp\left(\left(E-h\nu_{s}\right)/k_{B}T\right)$, where the energy $k_{B}T$ = 25.7 meV, the energy E corresponds to the energy associated with the absorption edge (i.e zero phonon energy) and $h\nu_{s}$ is the signal transition energy. Thus, the McAmber theory yields coefficient of $\sigma^{e}/\sigma^{a}$ = 5.6. In this way, a value of $\sigma^{a}$ = (5 ± 2) × 10^−14^ cm^2^ is the best fitting parameter in the calculated curve in Figure 4d (green continuous line) that is in good agreement with that reported in literature, 2 × 10^−14^ cm^2^ [46].

However, although an optical gain model can reproduce our experimental data, there are no physical evidences of stimulated emission in the experimental results presented here. An optical gain process through the HC-NCF should be manifested in the observation of ASE in PL spectrum under sufficiently high power pumping conditions, whose physical evidence is the collapse of the PL spectrum (around 20 nm wide) in Figure 4a into a much narrower line (below 2 nm wide) under high excitation powers, and a superliner growth of its intensity above a certain threshold [47]. Despite a slight narrowing observed in the probe signal linewidth (inset in Figure 4d), no remarkable change was observed for the main PL band by increasing laser pumping power (see Figure 4a–b), hence none of above given conditions are fulfilled. That is, the observed enhancement of the probe signal can be phenomenologically explained by a stimulated emission process, even if the pumping laser power coupled to PNCs is below the ASE threshold. Other possible gain assisted mechanisms include the superposition of the PL-beam produced by the laser pump and the probe signal or scattered light. However, the former is discarded by the deconvolution performed in the analysis of recorded spectra at the exit of the fiber, while the latter would follow a linear dependence on the laser pump power (i.e., no gain).

Alternatively, third-order nonlinear effects in PNCs may also explain the experimental results presented here, at least to be considered as another phenomenological model until being confirmed by other different experiments. In previous works, we demonstrated that similar pumping conditions can change the effective refractive index of colloidal QDs integrated in plasmonic [44] or dielectric waveguides [48] due to a nonlinear interaction of pump plus probe beams with QDs. On the other hand, a Four Wave Mixing (FWM) third-order nonlinear process has been commonly proposed to provide parametric gain in Fiber-based optical devices [49] or the generation of high energy femtosecond pulses [50]. In nonlinear optical media, the interaction between two incident beams (*E_1_(ω_1_)* and *E_2_(ω_2_)*) can generate two waves at different frequencies (*E_1_(ω_1_) + E_2_(ω_2_)* → *E_s_(ω_s_) + E_i_(ω_i_)*) if the phase matching condition (*ω_1_ + ω_2_ = ω_s_ + ω_i_*) is fulfilled. If *E_1_(ω_1_) = E_2_(ω_2_) = E_p_(ω_P_)* the process is called non-degenerate FWM and can be exploited to generate some kind of optical parametric gain at the created wavelengths, ω_s_ and ω_i_, when ω_P_ is tuned. In these conditions, the parametric gain generated by this process can be described by the following equation if the operative range of wavelengths (the difference between the pump and the signal wavelengths) does not exceed a band of 100 nm [51]:(5)$$G_{dB}=10\;\log_{10}\left(1+{\left[\frac{\gamma P_{a}}{g}\sin h\left(gL\right)\right]}^{2}\right)$$
where *G* is the normalized output signal (*G* = *P*_out_/*P*_in_) produced though a length *L*; the factor $\gamma =2\pi n_{2}/\lambda A_{\mathrm{eff}}$ is the effective nonlinear optical coefficient, where n_2_ is the fiber nonlinear parameter. The parametric gain coefficient *g* is given by:(6)$$g^{2}=-\Delta \beta \left(\frac{\Delta \beta }{4}+\gamma P_{p}\right)$$
where the phase mismatch ∆*β* is given by:(7)$$\Delta \beta =\beta \left(\omega_{s}\right)+\beta \left(\omega_{i}\right)-2\beta \left(\omega_{p}\right)$$

Under resonant conditions, Δβ≈0 and *g* ≈ 0. If we calculate the first terms of the Taylor expansion for low gL values, it follows that:(8)$$G_{dB}=10\log_{10}\left(1+{\left(\gamma P_{p}L\right)}^{2}{\left[1+{\frac{gL}{6}}^{2}+{\frac{gL}{120}}^{4}+\cdots \right]}^{2}\right)\approx 10\;\log_{10}\left(1+{\left(\gamma P_{p}L\right)}^{2}\right)$$

In this way, the experimental data in Figure 4d can also be nicely fitted to Equation (8) (orange continuous line) with $n_{2}$ = 3.8 × 10^−10^ m^2^/W and phase mismatch values Δβ<1°. Therefore, the model suggests that a material with a large Kerr effect would influence the propagation of a probe signal beam. Since the reported values of the silica nonlinear refractive index are limited to 10^−20^ m^2^/W under nanosecond excitation, we believe that nonlinear optical properties of PNCs can be responsible of the mechanism proposed here. Indeed, similar measurements with an unloaded HC-NCF did not show any signal enhancement. Although the application of PNCs in optical nonlinear devices is currently at its very beginning, the few experimental studies show promising properties [52]. The values reported for colloidal solutions of CsPbBr_3_ materials are about 2 × 10^−17^ m^2^/W [52] and 1.2 × 10^−16^ m^2^/W [53] for femtosecond and nanosecond excitation, respectively, but it is necessary to take into account that the concentration of PNCs in such colloidal solutions is usually low. Therefore, the nonlinear optical parameters can be much higher in thin films, as it is in our device. In any case, further research is needed to elucidate the ultimate origin of the nonlinear optical properties of PNCs under CW pumping and if they are sufficient to support a parametric gain mechanism. For the moment, this model can be considered as a phenomenological approach.

## 5. Conclusions

We present a novel HC-NCF optical amplifier by doping with highly luminescent CsPbBr_3_ nanocrystals prepared by chemical synthesis. The PNCs were infiltrated by capillary forces producing a 55 nm thick layer on the air tube walls of the fiber, producing a change in the optical modes at the wavelengths of emitted light by PNCs (510–540 nm), whereas the pumping laser was guided in the main optical mode of the HC-NCF (central hollow-core). A gain figure of merit of 3 dB was obtained for a 140 mm long fiber. Given the absence of ASE in CsPbBr_3_ PNCs under continuous wave laser pumping at room temperature, the observed amplification by the PNC-doped HC-NCF can be alternatively explained by means of a third-order nonlinear optical mechanism. This kind of promising device is expected to be a source of new and interesting applications in high power optical fiber lasers and optical telecommunications, and, more importantly, a workbench for the study of high-order optical nonlinear optics, combining the light confinement properties of HC-NCFs and the optical properties of the perovskite nanocrystals, which would be of great importance for both fundamental physics investigation and practical multiphoton applications. For example, more appropriate hollow-core optical fibers can be designed to increase the effective area of the tube modes and hence the light–matter interaction inside the fiber, as the basis to study high-order optical nonlinearities of perovskite nanomaterials and the development of new future nonlinear photonic devices.

## Figures and Tables

**Figure 1 nanomaterials-09-00868-f001:**
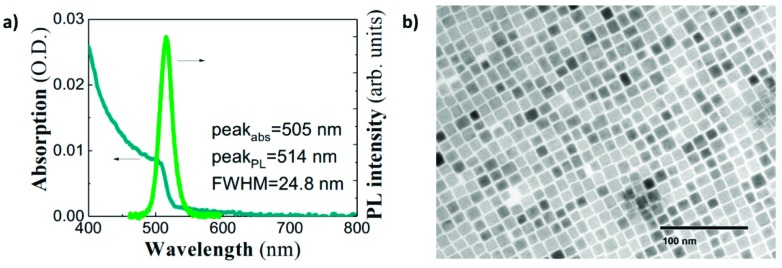
(**a**) Absorbance (blue continuous line) and PL (green continuous line) spectra of CsPbBr_3_ PNCs dispersed in hexane; and (**b**) TEM image of cubic CsPbBr_3_ PNCs. The average cube side is around 10 nm.

**Figure 2 nanomaterials-09-00868-f002:**
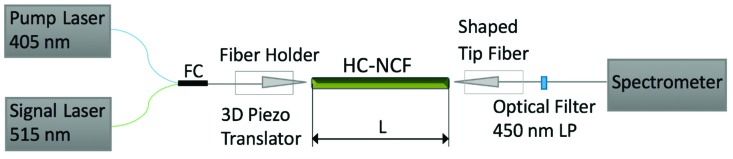
Experimental setup for optical gain measurements, consisting of two tip-shaped fibers coupled at both ends of the HC-NCF facets. CW-pump (405 nm) and probe signal (515 nm) beams are combined inside the first fiber by a fused fiber coupler (FC), whereas the second fiber collects light at the output of the HC-NCF into a spectrometer after a 450 nm long-pass filter.

**Figure 3 nanomaterials-09-00868-f003:**
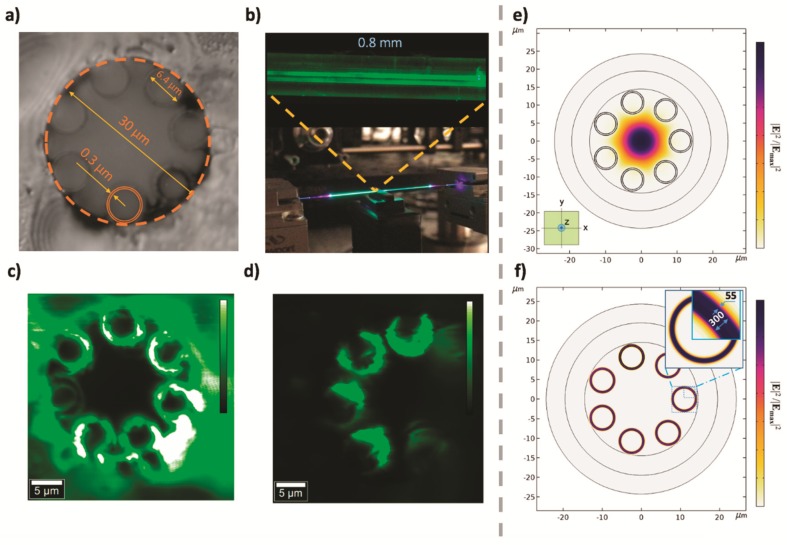
(**a**) Cross section of an undoped fiber taken with optical microscopy; (**b**) picture of the whole doped HC-NCF being pumped with the 405 nm CW laser and zoomed axial fragment (this image was registered with an optical microscope that assists the experimental setup described in Figure 2). (**c**,**d**) PL cross-section images at input and intermediate (after cleaving) faces of the HC-NCF, respectively; (**e**) simulation of the modal field distribution of the seven-tube HC-NCF fiber operating at a wavelength of 515 nm in the absence of PNCs. Z-axis corresponds to the propagation direction; and (**f**) modal field distribution of the HC-NCF doped with PNCs concentrated on the tube walls after considering a PNC layer with 55 nm thickness.

**Figure 4 nanomaterials-09-00868-f004:**
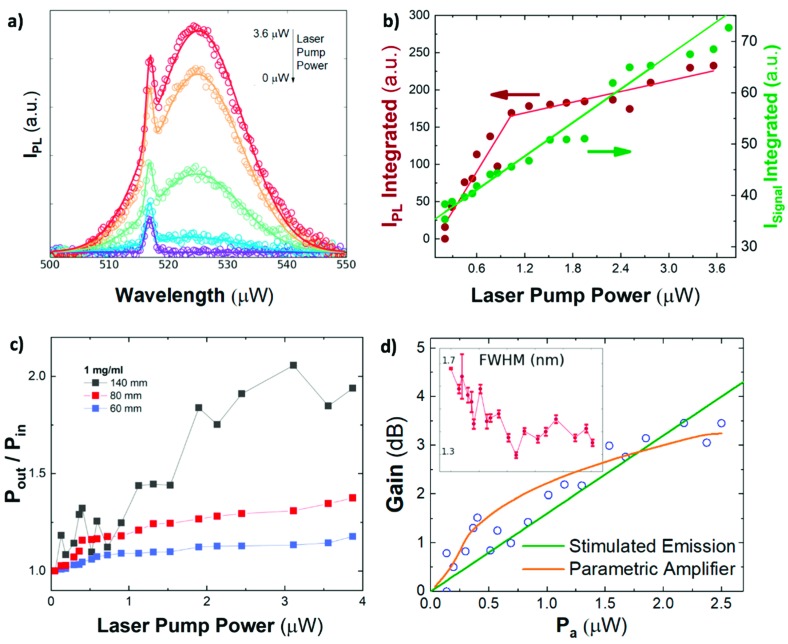
(**a**) Spectra recorded at the extreme of a 140 mm long HC-NCF filled with PNCs (concentration of 1 mg/mL) for different laser pumping powers. Symbols correspond to the experimental data and solid lines to the fitting by the sum of two Gaussian lineshapes (the PL of the PNCs and the regenerated 515 nm laser line, or probe). (**b**) Integrated intensity of the PL (brown circles) and probe (green circles) signals deconvoluted from the spectrum in (**a**) for the 140 mm long HC-NCF. (**c**) Optical amplification ratio *P_out_/P_in_* for different HC-NCF lengths, as a function of the laser pumping power. (**d**) Gain in dB as a function of the absorbed laser power in the 140 mm long HC-NCF doped with colloidal PNCs. Continuous curves stand for calculated ASE gain (green line) and parametric amplification (orange line). The inset shows the FWHM of the probe signal with error bars as extracted from fitting experimental curves in (**a**) (the x-axis range is the same as in the main figure).

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
