# Peer review of "Optical Amplification in Hollow-Core Negative-Curvature Fibers Doped with Perovskite CsPbBr3 Nanocrystals"

_nanomaterials, 2019, doi:10.3390/nano9060868_

Reviewer 1 Report

Nowadays the “revolver” structure of hollow core anti-resonant fiber is popular, hence presented results are very interesting. Generally, the article is clearly written and I recommend to publish this article after my few remarks:

I am not sure that is good idea to characterize this structure towards optical amplifiers. The power of signal range is 1nW and pumping power is up to 3uW. In this situation is difficult to find practical application such structures in optical amplification in the visible range. It is better to focus on sensing or nonlinearity behavior.

In Figure 2. the fiber is assigned by HC-ARF but in whole article Authors used HC-NCF. It should be normalized.

According to photos (fig3) I supposed that the homogeneity of PNCs in low(bright luminescence in right down side of structure)? How Authors confirmed the constant thickness of PNCs layer?

Also, I agree with Authors that stimulated emission does not exist in this material (line 264), and probably the signal amplification is connected with the superposition of pump and probe emission lines or even form scattering?

Author Response

See

Reviewer 2 Report

In this paper, the authors introduced a hollow-core negative-curvature fiber (HC-NCF) optical signal amplifier by 23 filling the air microchannels of the fiber with all-inorganic CsPbBr3 perovskite nanocrystals (PNCs).  The optimum fabrication conditions are found to enhance the optical gain, up to +3 dB in the best device. Experimental results were approximately reproduced by a gain assisted mechanism based 26 on the nonlinear optical properties of the PNCs, indicating that signal regeneration can be achieved under low pump powers, much below the threshold of stimulated emission. The results can pave the road of new functionalities of the HC-NCF with PNCs, such as optical amplification or nonlinear 29 frequency conversion. The idea behind this is interesting. However, I still have quite a number of concerns in this manuscript. There are times where there are not enough data to support the conclusions of the author. Please see some of the major concerns below.

1.The information for the HC-NCF structure is not enough. The authors should give much more information about this, where is the x-z plane and x-y plane ,So the readers can get its reproducibility. 

2.  The authors should give much more information about the novelty of this paper, especially the effect of using this structure as sensor, which applications can be used this device?

3. More references need to be included in the introduction part to understand the applications of using PCF and HPCF

1.      Prospects for diode-pumped alkali-atom-based hollow-core photonic-crystal fiber lasers"

Optics Letters, 39(16), 2014 (4655-4658)

2.      "Power splitting of 1x16 in multicore photonic crystal fibers"

Applied Surface Science, 417, 2017 (34-39)

3.      "An eight-channel C-band demux based on multicore photonic crystal fiber"

 Nanomaterials, 8(10), 2018, 845 (10 pages)

4.  Much more discussion about the results should be given in this paper, especially the author needs to provide enough physicals mechanism analysis about the results.

Author Response

The reply includes for both Referee's 1 and 2.

Round  2

Reviewer 2 Report

The modified  paper can be published